# Coexistence of Ehlers–Danlos Syndrome with Coronary–Pulmonary Arterial Fistula and Other Multiple Coronary Artery Anomalies

**DOI:** 10.3390/diagnostics13233555

**Published:** 2023-11-28

**Authors:** Paweł Gać, Arkadiusz Jaworski, Jakub Karwacki, Michał Jarocki, Artur Ams, Rafał Poręba

**Affiliations:** 1Centre for Diagnostic Imaging, 4th Military Hospital, 50-981 Wroclaw, Poland; 2Division of Environmental Health and Occupational Medicine, Department of Population Health, Wroclaw Medical University, 50-345 Wroclaw, Poland; 3Students’ Association of Psychiatry, Department of Psychiatry, Wroclaw Medical University, 50-367 Wroclaw, Poland; arkadiusz.jaworski.lek@gmail.com (A.J.); artureo058@gmail.com (A.A.); 4Department of Minimally Invasive and Robotic Urology, University Center of Excellence in Urology, Wroclaw Medical University, 50-556 Wroclaw, Poland; jkarwacki.md@gmail.com; 5Students’ Scientific Organization, Institute of Heart Diseases, Wroclaw Medical University, 50-556 Wroclaw, Poland; m.jarocki262@gmail.com; 6Department of Internal and Occupational Diseases, Hypertension and Clinical Oncology, Wroclaw Medical University, 50-556 Wroclaw, Poland; rafal.poreba@umw.edu.pl

**Keywords:** Ehlers–Danlos syndrome, coronary computed tomography angiography, CCTA, coronary–pulmonary arterial fistula, CPAF, coronary artery anomalies, ectopic coronary artery

## Abstract

This case report presents a 34-year-old male with Ehlers–Danlos syndrome, type 2 diabetes mellitus, aortic valve regurgitation, and aortic bulb aneurysm. Following spine surgery for thoracic–lumbar stabilization, the patient underwent assessment for aortic bulb aneurysm and aortic valve replacement surgeries. Five months post spinal surgery, a coronary computed tomography angiography was performed. The coronary computed tomography angiography revealed unique findings, including the absence of the left main coronary artery, right coronary artery dominance, ectopic origin of the left circumflex artery from the right sinus of the valsalva, a coronary–pulmonary arterial fistula originating from the right sinus of the valsalva, and an additional right pulmonary vein. The patient was qualified for surgical treatment for an aortic bulb aneurysm, was informed about the high surgical risk, and is awaiting surgery. This case underscores the rarity of Ehlers–Danlos syndrome coexisting with multiple coronary artery anomalies. The presence of a coronary–pulmonary arterial fistula further emphasizes the need for specialized patient monitoring when Ehlers–Danlos syndrome and coronary anomalies converge.

Ehlers–Danlos syndrome (EDS) encompasses a spectrum of heterogeneous, heritable disorders affecting connective tissue. According to the 2017 international classification of EDS, there are 13 subtypes. A rare 14th subtype was found in 2018. Each subtype has unique aspects, and a genetic cause has been identified in all subtypes except hypermobile EDS [1]. Clinical manifestations arise from the pervasive fragility and weakness of the soft connective tissue. Apart from manifestations in the skin, ligaments, joint and internal organs, there are subtypes of EDS which affect blood vessels and cardiac valves. These subtypes confer a higher risk of cardiac valve insufficiency, aneurysms, dissections, or ruptures of arteries, including coronary ruptures [2]. The estimated prevalence of EDS approximates 194.2 cases per 100,000 individuals [3]. Here, we present a case of a 34-year-old man with normal body weight, with hypermobile EDS, type 2 diabetes mellitus, chest deformity, aortic valve regurgitation, and aortic bulb aneurysm characterized by an axial diameter measuring approximately 5 cm (Figure 1). The patient had severe rotational scoliosis of the Th-L spine, with a significant deepening of the thoracic kyphosis and lumbar lordosis. There was a decrease in the AP dimension of the chest, a deformation of the ribs with a decrease in the width of some intercostal spaces, and a secondary widening of the width of other intercostal spaces. The patient underwent elective spine surgery to correct severe rotational scoliosis of the T-L spine. After postoperative rehabilitation, the patient underwent a thorough assessment of eligibility for aortic bulb aneurysm and aortic valve replacement surgeries. Consequently, a coronary computed tomography angiography (CCTA) was performed five months after the spinal surgical intervention. The CCTA revealed an absence of the left main (LM) and the right coronary artery (RCA) dominance. Significantly, the left circumflex artery (LCx) exhibited an ectopic origin from the right sinus of the valsalva, demonstrating a retroaortic course in the proximal segment and a typical course in the distal segments (Figure 2). Furthermore, the presence of a coronary–pulmonary arterial fistula (CPAF) in the CCTA was visualized. A vessel with an approximate diameter of 0.3 cm originated from the right sinus of the valsalva and progressed anteriorly and then towards the left, traversing anteriorly to the right ventricular outflow tract. Thereafter, this vessel was visible to the left of the main pulmonary artery, ultimately divided into multiple narrow segmentally aneurysmally dilated and tortuous fistula vessels. The fistula vessels showed communication with the proximal segment of the main pulmonary artery (Figure 3). Moreover, the imaging revealed the presence of an additional right pulmonary vein (Figure 4). The CCTA revealed calcifications within the aortic valve. The patient received a qualification card for surgical treatment, was informed about the high surgical risk, and is awaiting the operation.

To our knowledge, the coexistence of EDS with developmental anomalies of the coronary arteries remains ultra-rare. One paper mentions a patient with coexistent EDS and CPAF [4]. Coronary anomalies are mostly detected incidentally during coronary angiography. An absence of the LMS is the most common coronary anomaly, with an incidence ranging between 0.41 and 0.67%. An ectopic origin of the LCx from the right sinus of the valsalva or from the proximal RCA is the second-most-common coronary artery anomaly, with a frequency of 0.37% [5]. The prevalence of CPAF spans from 0.32% to 0.68%, and usually has no clinical significance. Nevertheless, it bears the potential for complications encompassing myocardial ischemia, pulmonary hypertension, congestive heart failure, and the development of coronary aneurysms [6]. Owing to factors including the presence of aneurysmally dilated and tortuous vessels within the framework of the CPAF, the concurrent existence of these anomalies with EDS necessitates specialized patient monitoring. The causal role of EDS in the occurrence of this type of anomaly should also be further investigated.

## Figures and Tables

**Figure 1 diagnostics-13-03555-f001:**
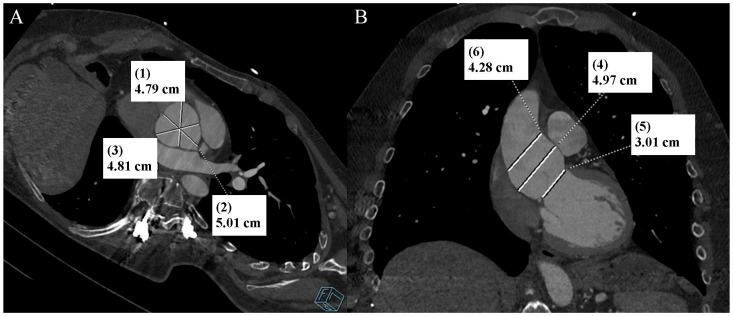
Thoracic aorta computed tomography angiography. Aortic bulb aneurysm. (**A**). Multiplanar reconstruction. View parallel to the aortic annulus plane. Subsequent measurements were made between the commissures of the aortic valve leaflets and the midpoints of the aortic valve leaflets. (**B**). Multiplanar reconstruction. Perpendicular view to the aortic annulus plane in the long axis. The designations (1–6) indicate subsequent measurements: aortic annulus—3.01 cm (5); aortic bulb—4.79 cm (1), 5.01 cm (2), 4.81 cm (3), 4.97 cm (4); sinotubular junction—4.28 cm (6).

**Figure 2 diagnostics-13-03555-f002:**
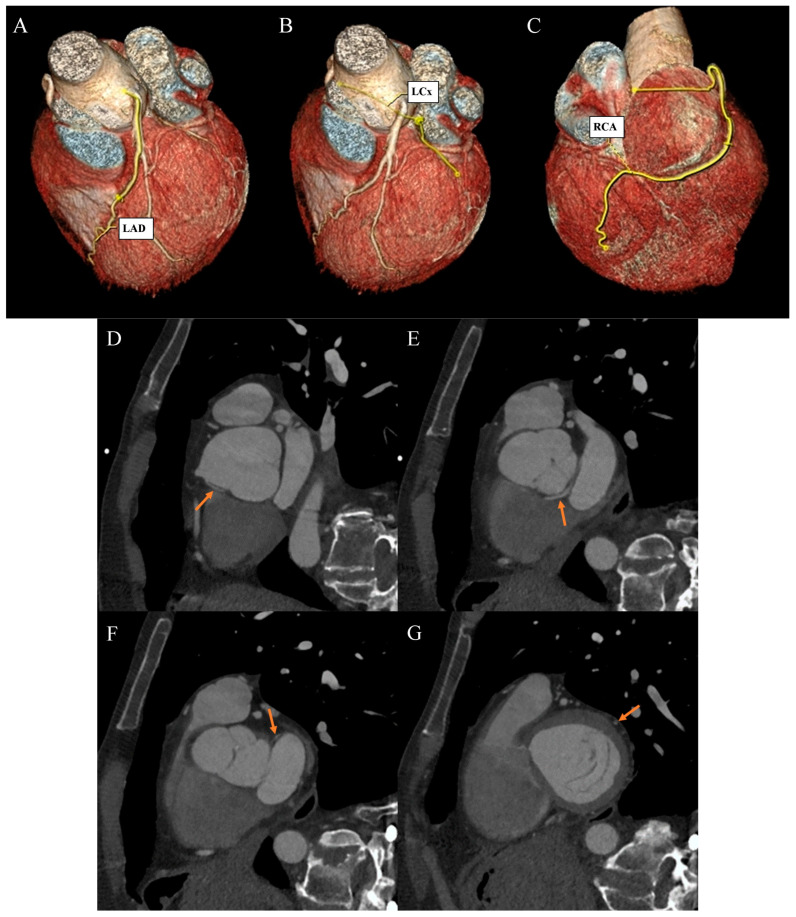
Coronary artery computed tomography angiography. Coronary artery anomaly. (**A**) Volume-rendering technique reconstruction. Left anterior descending (LAD) artery with a direct origin from the left aortic bulb sinus. (**B**) Volume-rendering technique reconstruction. Distal left circumflex artery (LCx) segment as a continuation of the branch located between the aortic bulb and left atrium wall. (**C**) Volume-rendering technique reconstruction. Dominant right coronary artery (RCA) with a typical course. (**D**) Multiplanar reconstruction. Oblique view. The arrow indicates the ectopic origin of LCx from the right aortic bulb sinus. (**E**) Multiplanar reconstruction. Oblique view. The arrow indicates the retroaortic course of the proximal LCx segment. (**F**) Multiplanar reconstruction. Oblique view. The arrow indicates the course of the medial LCx segment between the aortic bulb and left atrium. (**G**) Multiplanar reconstruction. Oblique view. The arrow indicates a typical course of the distal LCx segment.

**Figure 3 diagnostics-13-03555-f003:**
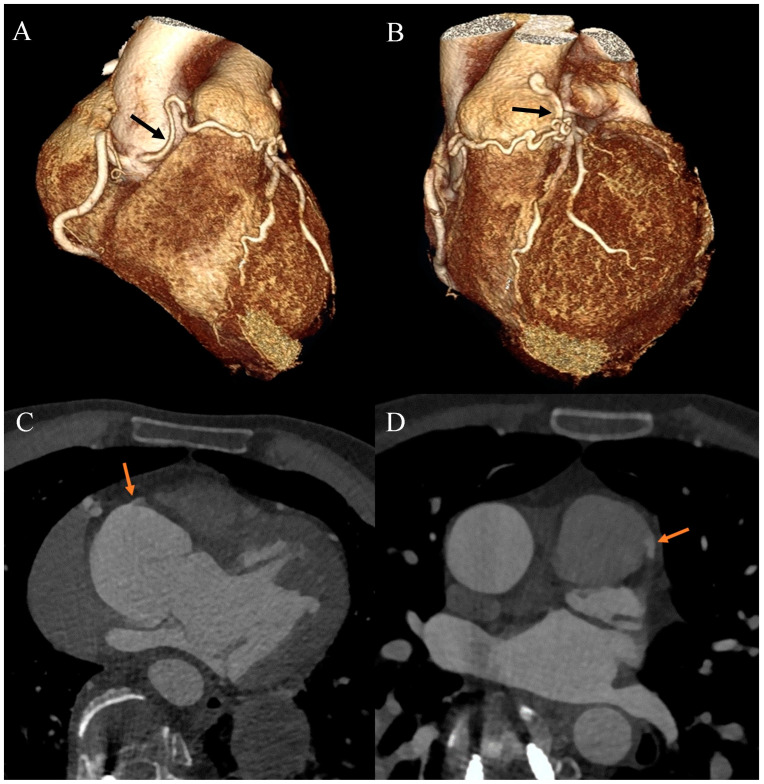
Coronary artery computed tomography angiography. Coronary–pulmonary arterial fistula. (**A**) Volume-rendering technique reconstruction. Anterior-right view. The arrow indicates the proximal part of the fistula. (**B**) Volume-rendering technique reconstruction. Anterior-left view. The arrow indicates the distal part of the fistula. (**C**) Multiplanar reconstruction. Axial view. The arrow indicates the origin of the fistula from the right aortic bulb sinus. (**D**) Multiplanar reconstruction. Axial view. The arrow indicates the connection between the fistula and the main pulmonary artery.

**Figure 4 diagnostics-13-03555-f004:**
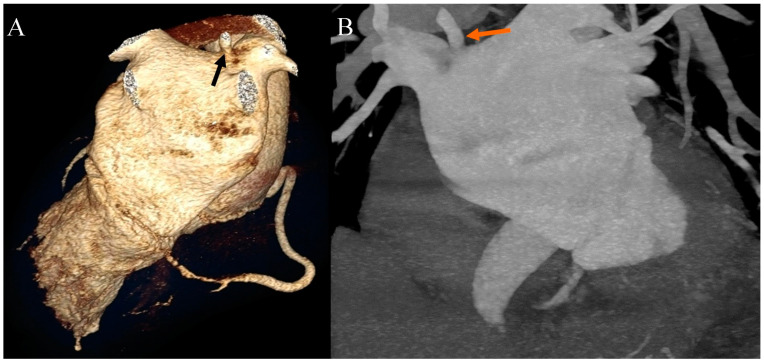
Coronary artery computed tomography angiography. Accessory right pulmonary vein. (**A**) Volume-rendering technique reconstruction. Posterior view. The arrow indicates the accessory right pulmonary vein on the upper wall of the left atrium. (**B**) Maximum intensity projection reconstruction. Anterior view. The arrow indicates the accessory right pulmonary vein on the upper wall of the left atrium.

## Data Availability

No new data were created or analyzed in this study.

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
