# Peer review of "Coexistence of Ehlers–Danlos Syndrome with Coronary–Pulmonary Arterial Fistula and Other Multiple Coronary Artery Anomalies"

_diagnostics, 2023, doi:10.3390/diagnostics13233555_

Round 1
Reviewer 1 Report
Comments and Suggestions for Authors
In the manuscript titled ’Concurrent Presence of Ehlers-Danlos Syndrome with Coronary-Pulmonary Arterial Fistula and Multiple Coronary Artery Anomalies’, Paweł Gać and colleagues presented a case involving a 34-year-old man diagnosed with Ehlers-Danlos syndrome (EDS), aortic valve regurgitation, aortic bulb aneurysm, and multiple coronary artery anomalies. The authors commence with a clinical case presentation, followed by a concise overview. The case is enriched with a multitude of high-quality images. The title accurately reflects the manuscript's content, the abstract is comprehensive and summarizes key details. The case is excellently composed and holds significant educational value within the realm of medical education.
I have the following comments:
1. There are unnecessary abbreviations in the abstract: LM, RCA, LCx. Since these abbreviations are not used later in the abstract, it is not necessary to provide their abbreviations here.
2. The introductory section (lines 31-37) provides an overview of the fundamental aspects of EDS. Nonetheless, it appears somewhat superficial, and there is room for the authors to delve deeper into certain aspects, particularly pertaining to the primary types of EDS and their clinical presentations.
3. In the case presentation, it would be beneficial to specify the exact subtype of EDS the patient had and provide a detailed description of the specific chest deformity they experienced.
4. It would also be of interest to readers to know why the spinal surgery was necessary.
5. In the case of Figures, it would provide a better visual experience if the alignment and size of sub-images within a single image were consistent.
6. In Figure 1 caption, it would be helpful to include the numbers seen within square brackets in the images for easier comprehension. I recommend using the full term instead of the abbreviation ‘STJ’.
7. In Figures 2 A-C, the text appears pixelated.
8. I suggest maintaining consistency in the abbreviation for the circumflex artery, as it has been referenced as LCx, Cx, and CX.
Comments on the Quality of English Language
Spell checking for minor errors is recommended, and using appropriate synonyms instead of frequently repeated words is advised.
Author Response
Dear Reviewer,
Please see the attachment.
Best regards,
Authors

Reviewer 2 Report
Comments and Suggestions for Authors
Comments on the Quality of English LanguageAuthor Response
Dear Reviewer,
Please see the attachment.
Best regards,
Authors

Reviewer 3 Report
Comments and Suggestions for Authors
The paper is interesting.
Line 18: EDS is reported for the first time as an abbreviation in the Abstract. It should be indicated in clear words:" Ehlers-Danlos Syndro,me (EDS)".
The Authors, both in the Abstract and in the text, should also report the clinical decisions which have been made in this case and the results of the follow-up and its lenghth in time.
In the conclusions the Authors could add that the possibility of a causal role of EDS for the occurrence of these anomalies should be further explored.
Comments on the Quality of English LanguageMinor editing required.
Author Response

(The authors gave the same response as above.)

Reviewer 4 Report
Comments and Suggestions for Authors
The authors describe a case report of a 34-year-old male with EDS, type 2 diabetes, aortic valve regurgitation, and aortic bulb aneurysm.
- They describe very uncommon incidental findings
- The quality of images CT scan is good and appropriate for this case report
- They acknowledge the rarity of the comorbidity with EDS and the possible clinical insignificance of the findings (despite in rare cases could lead to myocardial ischemia, pulmonary hypertension, congestive heart failure, and the development of coronary aneurysms).
As this finding is rare, in my opinion this case report worth to be published.
Author Response

(The authors gave the same response as above.)

Reviewer 5 Report
Comments and Suggestions for Authors
This is an adequaetly described case report. I suggested changing the term ''type 2 diabetes'' to ''type 2 diabetes mellitus'' and to avoid/replace the abbreviations in the abstract.
Author Response

(The authors gave the same response as above.)
